# Acute Colon Inflammation Triggers Primary Motor Cortex Glial Activation, Neuroinflammation, Neuronal Hyperexcitability, and Motor Coordination Deficits

**DOI:** 10.3390/ijms23105347

**Published:** 2022-05-11

**Authors:** Livia Carrascal, María D. Vázquez-Carretero, Pablo García-Miranda, Ángela Fontán-Lozano, María L. Calonge, Anunciación A. Ilundáin, Carmen Castro, Pedro Nunez-Abades, María J. Peral

**Affiliations:** 1Departamento de Fisiología, Facultad de Farmacia, Universidad de Sevilla, 41012 Sevilla, Spain; livia@us.es (L.C.); mvazquez1@us.es (M.D.V.-C.); calonge@us.es (M.L.C.); ilundain@us.es (A.A.I.); pnunez@us.es (P.N.-A.); 2Instituto de Biomedicina de Cádiz (INIBICA), 11009 Cádiz, Spain; Carmen.castro@uca.es; 3Departamento de Fisiología, Facultad de Biología, Universidad de Sevilla, 41012 Sevilla, Spain; afontan@us.es; 4Departamento de Fisiología, Facultad de Medicina, Universidad de Cádiz, 11003 Cádiz, Spain

**Keywords:** colon inflammation, motor neurons, microglial and astrocyte activation, neuroinflammation, neurodegeneration, hyperexcitability, motor coordination

## Abstract

Neuroinflammation underlies neurodegenerative diseases. Herein, we test whether acute colon inflammation activates microglia and astrocytes, induces neuroinflammation, disturbs neuron intrinsic electrical properties in the primary motor cortex, and alters motor behaviors. We used a rat model of acute colon inflammation induced by dextran sulfate sodium. Inflammatory mediators and microglial activation were assessed in the primary motor cortex by PCR and immunofluorescence assays. Electrophysiological properties of the motor cortex neurons were determined by whole-cell patch-clamp recordings. Motor behaviors were examined using open-field and rotarod tests. We show that the primary motor cortex of rats with acute colon inflammation exhibited microglial and astrocyte activation and increased mRNA abundance of interleukin-6, tumor necrosis factor-alpha, and both inducible and neuronal nitric oxide synthases. These changes were accompanied by a reduction in resting membrane potential and rheobase and increased input resistance and action potential frequency, indicating motor neuron hyperexcitability. In addition, locomotion and motor coordination were impaired. In conclusion, acute colon inflammation induces motor cortex microglial and astrocyte activation and inflammation, which led to neurons’ hyperexcitability and reduced motor coordination performance. The described disturbances resembled some of the early features found in amyotrophic lateral sclerosis patients and animal models, suggesting that colon inflammation might be a risk factor for developing this disease.

## 1. Introduction

Neuroinflammation is considered an important inducer for the development of neurodegenerative diseases, such as Parkinson’s, Alzheimer’s, and amyotrophic lateral sclerosis (ALS) [1]. Astrocytes and microglia, which exhibit either resting or activated states depending on the surrounding environment, play a key role in neuroinflammation. Reactive astrocytes exhibit increased size and release inflammatory mediators [1]. Activated microglia display increased body size and release pro-inflammatory cytokines, such as interleukin-1 beta (IL-1β), interleukin-6 (IL-6), and tumor necrosis factor-alpha (TNF-α) [2]. Overactive microglia and astrocytes also release reactive oxygen species, such as nitric oxide (NO), that primarily results from increased expression of the inducible nitric oxide synthase (iNOS) [2]. Another form of NOS is present exclusively in neurons (nNOS) [3]. The released pro-inflammatory cytokines and NO, in turn, activate surrounding glial cells, inducing a self-potentiating cycle that maintains a pro-inflammatory environment [2].

Central nervous system (CNS) inflammation can be elicited by peripheral inflammation [4]. Diseases that involve peripheral inflammation are the inflammatory bowel diseases (IBD), such as ulcerative colitis. Patients with these disorders exhibit intestinal symptoms such as diarrhea and rectal bleeding, but also often suffer from cognitive and neuropsychiatric disfunctions [5,6]. In fact, the intestinal inflammation has been considered a risk factor for suffering from Parkinson’s, Alzheimer’s, and ALS diseases [7,8], and the extent of the intestinal disorders observed in these patients are such that they are considered part of their clinical picture [9]. Animals with experimental colitis display behavior and morphological brain alterations similar to those of IBD patients, as well as inflammation in some brain regions [10,11]. In these animals, the neuroinflammation is accompanied by microglial activation in the hippocampus, cerebral cortex, and mesencephalon [11,12,13,14,15,16], increased excitability in the hippocampus and dorsal root ganglia [12,17,18], and reduced locomotor activity [11,19,20]. Together, these observations indicate that colitis animal models are suitable for studying the gut–brain interactions [10].

The gut–brain axis is a bidirectional link between the CNS and the gut, providing an explanation of how changes in the gut barrier affect brain homeostasis, and vice versa [8,10]. As far as we know, however, no study has addressed the effect of acute colon inflammation on the primary motor cortex. ALS disease is characterized by motor neurons’ degeneration, including pyramidal neurons of the primary motor cortex (layer V) that regulate voluntary motor output [21]. In most cases, ALS is a sporadic disease and the initial cause(s) remain(s) unknown, but besides activated microglia and astrocytes, and neuroinflammation [22,23], ALS patients and animal disease models exhibit motor cortical hyperexcitability that seems to precede the neurodegeneration associated with ALS [24]. What produces enhanced excitability and makes the pyramidal neurons of the primary motor cortex vulnerable to degeneration is not completely understood. 

Accumulating evidence suggests that one possible cause for the neuronal hyperexcitability may be the direct action of inflammatory and/or oxidative stress mediators (cytokines and NO) on both neuron membrane intrinsic properties and synapsis, which, in the long term, may contribute to neuronal injury and death [25,26,27].

The aim of the present study was to determine whether acute colon inflammation causes microglial and astrocyte activation and neuroinflammation in the primary motor cortex and alters motor neuron functions. To achieve this, we used a well-established rat model of acute colitis and evaluated both the intrinsic electrical properties of motor cortex neurons and the animal motor coordination. The study might contribute to understanding the role played by inflammation in the vulnerability of cortical motor neurons to degeneration and death, feature symptoms of the ALS pathology. 

## 2. Results

### 2.1. DSS Treatment and Colon Inflammation

The induction and progression of colon inflammation were evaluated by determining the disease activity index (DAI) and colon parameters, as described in the Methods Section. All the rats exposed to 3% DSS for 7 days developed acute colitis, as evidenced by the increase in the DAI, the shortening in the colon length, and the increase in the mRNA relative abundance of the pro-inflammatory cytokines IL-1β, IL-6, and TNF-α (Figure 1A–D). The loss of crypt architecture and the massive infiltration of inflammatory cells in the mucosa and submucosa (Figure 1E,F) are also indicative of colon inflammation.

### 2.2. Colon Inflammation Induces Neuroinflammation in Primary Motor Cortex

Once we established that 7 days of DSS treatment induces acute colonic inflammation in rats (Figure 1A–F), we tested whether these rats developed neuroinflammation by measuring the mRNA relative abundance of the pro-inflammatory mediators IL-1β, IL-6, TNF-α, iNOS, and nNOS. Total RNA was obtained from homogenates of the primary motor cortex of control (untreated) and DSS-treated rats. 

Figure 2A shows that, as compared with control rats, 7 days of DSS treatment significantly increased both IL-6 and TNF-α mRNA relative abundance by about 3- and 2-fold, respectively, with no change in that of IL-1β. An almost 4-fold increase in both iNOS and nNOS mRNAs was found in DSS-treated rats as compared with controls (Figure 2B).

These observations demonstrate that DSS treatment induced neuroinflammation in the primary motor cortex.

### 2.3. Colon Inflammation and Microglial and Astrocyte Activation in Primary Motor Cortex 

Activated microglial cells produce nitric oxide and undergo morphological changes characterized by an increase in their body size and retraction of their processes. To determine whether DSS-induced colon inflammation activates the primary motor cortex microglia, we quantified the microglial cell body size/cell size ratio by Iba1-immunostaining, as described in the Methods Section. The number of either Iba1, iNOS, or iNOS-Iba1 positive cells was also evaluated. 

DSS treatment significantly increased the microglial cell body/total cell size ratio, the number of iNOS^+^ cells (about 4-fold), and that of Iba1^+^ cells colocalizing with iNOS (about 7-fold), without modifying the number of Iba1^+^ cells (Figure 2C–G). The number of iNOS^+^ cells was greater than that of Iba1^+^-iNOS^+^ cells, indicating that non-microglial cells expressed iNOS and that both microglial and non-microglial cells responded to colon inflammation with iNOS induction.

Then, we examined whether DSS-induced colon inflammation also activates astrocytes in the primary motor cortex. For that, we performed a quantitative GFAP immunofluorescence assay, as described in the Methods Section. The results revealed a significant upregulation of GFAP expression in DSS-treated rats as compared with controls (Figure 3A,B). 

These findings, together with the observations summarized in Figure 2A,B, demonstrate that the DSS-induced acute colon inflammation triggers both neuroinflammation and activation of microglia and astrocytes in the primary motor cortex.

### 2.4. Colon Inflammation Depolarizes Membrane Potential and Increases Membrane Resistance of Motor Neurons 

We next determined whether DSS-induced colon inflammation affected the intrinsic electrical properties of the pyramidal neurons of the motor cortex. The whole-cell patch-clamp technique was applied. Recordings in current clamp mode were made in the pyramidal neurons of control (untreated) and DSS-treated rats, as described in the Methods Section. Figure 4A exemplifies a pyramidal neuron with a patch pipette on it.

Initially, we measured resting membrane potential and membrane input resistance (passive membrane properties). Figure 4B,C shows that at rest, the pyramidal neurons of both experimental groups did not discharge action potentials spontaneously, exhibiting a stable resting membrane potential that in DSS-treated rats was significantly more depolarized (Table 1). Figure 4B also illustrates the responses of representative pyramidal neurons to hyperpolarizing and depolarizing current pulses. The recordings revealed that the magnitudes of the membrane potential changes, induced by the current pulses, were higher in the DSS-treated rats, as compared with untreated rats. In addition, depolarizing currents of either 40 or 50 pA generated action potentials only in DSS-treated rats (Figure 4B). Figure 4D shows the voltage–current relationship of the neurons illustrated in Figure 3B. The input membrane resistance, obtained from the slope of this relationship, was significantly increased in DSS-treated rats as compared to control rats (Figure 4E; Table 1). 

### 2.5. Colon Inflammation Induces Hyperexcitability in Motor Neurons

Next, we looked at the active membrane properties of the pyramidal neurons from the primary motor cortex. In these experiments, we used whole-cell patch-clamp recordings in current clamp mode and measured, in control (untreated) and DSS-treated rats, rheobase, voltage threshold, depolarization voltage, action potential amplitude and duration, frequency gain, maximum frequency of discharge, and cancellation current. Observations made in DSS-treated rats were compared to those in control (untreated) rats.

As seen in a representative pyramidal neuron (Figure 5A) and in the whole neuron population studied in each experimental group (Figure 5B,C; Table 1), the rheobase was reduced by 50% and the depolarization voltage was reduced by 35% in DSS-treated rats compared to control rats. The voltage threshold value was similar in the two experimental groups (Figure 5D). As DSS treatment depolarized the resting membrane potential and did not affect the voltage threshold, the depolarization voltage value was decreased (Figure 5C; Table 1).

Compared to controls, DSS-treated rats did not show significant differences neither in the duration nor in the amplitude of the action potential, despite a tendency to decrease (Table 1). DSS-induced acute colon inflammation also modified the firing properties of pyramidal neurons from the primary motor cortex. Figure 6A shows that the motor neurons, from both experimental groups, responded to a depolarization current of 1 s duration by discharging repetitive action potentials, but with higher fire frequency in DSS-treated rats. The current–frequency relationship (Figure 6B) reveals that DSS treatment increased the frequency gain of the neurons by about 2-fold for the whole cell population (Figure 6C; Table 1). The maximum frequency achieved by the motor neurons, regardless of the current intensity, was not modified by DSS treatment (Figure 6D; Table 1), and the cancellation current in the neurons of DSS-treated rats was lower than in control rats (Figure 6E; Table 1). 

These alterations, along with those observed in the passive properties, indicate that the DSS-induced acute colitis neuroinflammation results in hyperexcitability of primary motor cortex neurons.

### 2.6. Colon Inflammation Causes Locomotion and Motor Coordination Deficits

As DSS-induced acute colon inflammation results in hyperexcitability of primary motor cortex neurons, we next studied motor behavior by evaluating locomotor activity and motor coordination in control (untreated) and DSS-treated rats. 

Spontaneous locomotor activity was evaluated at the end of the DSS treatment using an activity cage, as described in the Methods Section. We first measured horizontal activity for 10 min, including: total and per minute activity, total distance travelled, locomotion in both the center and the periphery of the arena, and the percentage of time spent within the center and the periphery of the arena (Figure 7; Table 2). Compared to controls, DSS-treated rats exhibited a significant decrease in both total horizontal motor activity and that monitored per minute (Figure 7A; Table 2). In addition, Figure 7A revealed that both experimental groups were affected by the time factor, showing a significant decline in motor activity with time, which indicates a correct habituation to the new environment. DSS treatment also significantly decreased the distance travelled (Figure 7B) and the locomotion either in the periphery or in the center of the arena (Figure 7C). No significant differences were observed between control and DSS-treated rats in the time spent in the center of the arena, which is an indirect anxiety-like behavior parameter (Figure 7D). 

Changes in motor behaviors that require coordination were also monitored by using the active cage. We measured the number of stereotyped movements, total vertical motor activity (expressed in number of rearings), maximum speed in centimeters/second, percentage of time spent resting, slow locomotion, and fast locomotion for 10 min (Figure 8A–D; Table 2). DSS-induced acute colitis reduced the number of stereotypes (Figure 7A) and rearings (Figure 8B), the maximum velocity of movement (Figure 8C), and the slow or fast locomotion (Figure 8D), whereas it increased the time spent resting (Figure 8D). To further confirm that DSS-treated rats displayed motor coordination defects, we assessed two trials of the rotarod test in a constant acceleration protocol (Figure 8E,F) and measured the latency to fall, in seconds, and the maximum speed, in rotations per minute (rpm). In both trials, DSS-treated rats showed significant decreases in permanence time on the rotarod test (Figure 8E) and in the maximum speed reached before the rats fell (Figure 8F). 

In addition to the results on the time spent in the center of the cage that indicate an absence of anxiety, DSS-treated rats did not display signs of pain, such as abnormal gait or posture, or excessive diminution of body weight gain, and muscular strength and grooming were also normal (Appendix A). These parameters indicate that it is unlikely that the observed motor impairments in DSS-treated rats were due to sickness behaviors. 

## 3. Discussion

Accumulating evidence supports that the peripheral inflammation, induced by inflammatory bowel diseases (IBD), generates neuroinflammation [4,7,8] which, in turn, triggers neurodegenerative diseases, such as ALS [28]. Herein, we showed that acute colon inflammation induces primary motor cortex microglial and astrocyte activation and neuroinflammation, together with pyramidal neurons’ hyperexcitability and motor coordination impairment. These alterations resemble those observed in ALS patients and animal models [22,24,29,30,31].

Inflammation development in the primary motor cortex of DSS-treated rats was indicated by the increased mRNA abundance of the pro-inflammatory cytokines, IL-6 and TNF-α, iNOS, and nNOS. Consistent with these results, the immunofluorescence assays revealed that the microglia of this brain region changed to a pro-inflammatory and cytotoxic phenotype. However, since Iba1 is a pan-microglial marker, we could not distinguish resident from peripheral microglia. The increase in mRNA levels of NOSs suggests augmented formation of NO, a mediator of both inflammatory and oxidative stress. These observations agree with those found in the hippocampus, cerebral cortex, and mesencephalon of animals with acute colitis, showing microglial activation and neuroinflammation with increases in pro-inflammatory cytokines IL-1β, IL-6, or TNF-α [11,12,13,14,15,16]. Rises in iNOS expression and protein nitration, indicative of an excessive NO production, were also observed in the circumventricular brain regions of those animals [11]. In addition, the increased GFAP staining in DSS-treated rats indicated astrocyte activation, as observed in the hippocampus [32]. The neuroinflammation could be triggered by the intestinal pro-inflammatory mediators that, reaching the brain via the blood circulation, induce production of inflammatory and neurotoxic mediators from microglia and astrocytes in a self-potentiating cycle, which in turn cause functional alterations or even neuronal degeneration [4,7]. The gut–brain connection might be responsible for the neuropsychiatric and cognitive dysfunctions, as well as the structural and functional alterations found in several brain areas, mainly in the limbic system, observed in IBD patients and animal models [10,11,33,34,35].

Our electrophysiology measurements revealed that the acute colon inflammation modified the intrinsic electrical properties of motor cortex pyramidal neurons, changes that led to neuronal hyperexcitability. Thus, the DSS-induced acute colitis depolarized the motor neuron membrane and increased its resistance at resting conditions. In these animals, lower depolarizing currents generated action potentials. Accordingly, the acute colitis-induced increase in membrane resistance together with the depolarized resting membrane potential produced a diminution of the rheobase, which may lead to increased excitability, as we previously described in motor cortex neurons of rats with induced oxidative stress [36,37]. The colon inflammation also increased both the action potentials’ firing frequency at any current intensity, that in part could result from the membrane depolarization which approaches the resting membrane potential to the unchanged excitation threshold, and the action potentials’ frequency gain, that along with the membrane depolarization, could trigger the motor neuronal hyperexcitability.

The neuronal hyperexcitability may be caused by a combination of an increase in inward cationic currents and/or a decrease in the outward currents. The membrane depolarization and the rise in its resistance, found in the DSS-treated rats, suggest alterations in membrane permeability, that could be due to changes in the ion conductance active at resting conditions. On the other hand, the increased action potentials’ frequency gain would involve modifications in the function and/or expression of voltage-gated ion channels. Chen et al. [18] observed that DSS-induced colitis increased the excitability of the dorsal root ganglion neurons, evidenced by the depolarized resting potential, decreased rheobase, and higher action potentials’ frequency. The authors attributed these effects to changes in ion channels’ expression, such as downregulation of voltage-gated K+ channels and upregulation of TRP (transient receptor potential) channels, that are inward cationic currents active at resting conditions [18]. In the case of the cortical motor neuron hyperexcitability observed in ALS patients and animal models, it is suggested to rise from increased Na+ currents and/or reduced K+ currents [38]. Cortical hyperexcitability may also result from an excitatory/inhibitory synaptic imbalance mediated by either GABA_A_ or glutamate receptors. In this sense, we have previously found that oxidative stress induced hyperexcitability in rat motor cortex neurons by blocking the GABA_A_ receptor-mediated inhibitory currents (active at resting conditions), leading to increased resistance and decreased rheobase. We have also seen that oculomotor nucleus neurons exposed to glutamate exhibited reduced rheobase and voltage depolarization and increased frequency gain [37,39,40,41]. Synaptic dysfunction that leads to hyperexcitability in ALS pathology is characterized by increased glutamatergic excitatory neurotransmission, which results from a combination of excessive neurotransmitter pre-synaptic release and inhibited reuptake [42]. Reduced motor cortical GABAergic inhibition has also been reported in ALS [24,42].

Growing evidence points out that excessive activation of neuronal IL-6 and TNF-α receptors may lead to hyperexcitability and excitotoxicity [27]. Our data revealed that the acute colon inflammation stimulated towards a microglial pro-inflammatory phenotype and increased the abundance of IL-6, TNF-α, iNOS, and nNOS mRNA in the primary motor cortex. Both cytokines, IL-6 and TNF-α, can increase voltage-gated Na+ channels’ activity [27]. IL-6 also promotes internalization of the GABA_A_ and glutamate receptors [27]. TNF-α induces glutamate release from microglia and astrocytes in an autocrine manner, and in neurons, mainly described in the hippocampus, decreases membrane insertion of GABAA receptors and increases that of glutamate [27,43]. Studies in animal models of acute colitis showed increased hippocampal excitability, that was dependent on microglial activation and mediated by TNF-α [12], together with increased excitatory glutamatergic transmission [17]. Increased NO production, generated by the action of iNOS and/or nNOS, could also contribute to the motor neuron hyperexcitability found here. Some reports showed that high levels of NO may induce hyperexcitability and kill neurons through inhibition of K+ resting currents and activation of TRP channels, both leading to neuronal depolarization [44,45]. Others found that NO increased glutamate release and affected the voltage- and/or ligand-gated ion channels [26,44]. Therefore, the induced motor neuron hyperexcitability found in our work could be due to a combined effect of the cytokines and NO that alter the balance between excitation and inhibition.

The current results also revealed that acute colon inflammation caused motor deficits. The lack of anxiety and pain signs, normal grooming and muscular strength, and not excessive diminution in body weight gain in DSS-treated rats indicates that the observed changes in motor activity were not due to sickness behaviors. In other studies, rats with DSS-induced colitis exhibited anxiety- and depression-like behaviors, but higher concentrations of DSS were used [18]. Our findings agree with previous studies showing diminished horizontal locomotor activity in animal models of acute colitis [19,20], but the present data are new in that they revealed more motor impairments, such as decreased stereotypes, vertical activity, maximum velocity, slow or fast locomotion, and motor coordination deficits evaluated with the rotarod test. Impaired mobility and muscle function are also exhibited by patients with ulcerative colitis [46,47,48].

The motor deficits described here indicate that the primary motor cortex has been altered functionally, probably due to the motor neurons’ hyperexcitability, and they resemble the early motor dysfunctions found in ALS animal models, such as the motor incoordination observed with the rotarod test [29,30,31]. At the early stages of the disease, ALS patients and animal models display functional modifications in the motor cortical networks, particularly in the primary motor cortex, which correlate with the early motor manifestations [24,49,50,51]. The motor cortical hyperexcitability observed in ALS is considered an early feature of the pathology and initiator of the pyramidal motor neurons’ degeneration, followed by degeneration of brainstem and spinal cord motor neurons [24,51]. 

## 4. Materials and Methods

### 4.1. Materials

The following antibodies and dilutions have been used: anti-ionized calcium-binding adapter molecule 1 (Iba1) (AbCam, Cambridge, UK; ab178846; 1:200), anti-iNOS (AbCam, Cambridge, UK; ab49999; 1:200), anti-glial fibrillary acidic protein (GFAP) (AbCam, Cambridge, UK; ab7260; 1:600), Alexa Fluor-488 (Invitrogen, Thermo Fisher, Madrid, Spain; A-11001; 1:100), and Alexa Fluor-546 (Life technologies, Thermo Fisher, Madrid, Spain; A-11010; 1:100). Unless otherwise indicated, the other reagents used in the study were from Sigma-Aldrich (Darmstadt, Germany).

### 4.2. Animals 

One-month-old male Wistar rats were housed in a 12:12 light/dark cycle and fed ad libitum with normal rodent diet and free access to tap water. They were humanely handled, and efforts were made to minimize the number of animals used and their suffering. The procedures concerning the protection of experimental animals were in accordance with the guidelines of the European Union Council (Directive 2010/63/UE) and the Spanish Royal Decree (BOE 34/11370, 2013). The research protocol was approved (15/01/2020/001, 23/01/2020) by the Animal Ethics Committee of the University of Seville and Junta de Andalucía. The rats were obtained from the “Centro de experimentación animal de la Universidad de Sevilla” located in the “Centro de Investigación Tecnología e Innovación de la Universidad de Sevilla III” (CITIUS III), Manuel Losada Villasante, Sevilla. 

### 4.3. Induction and Assessment of Experimental Colitis

Rats of similar body weight values were randomized into control (untreated) and DSS-treated groups, which received either drinking water or water containing 3% (wt/vol) dextran sulphate sodium (DSS, MW 40 kDa; TdB Consultancy, Uppsala, Sweden), respectively, over 7 days. At the end of the treatment, all animals were anesthetized by intraperitoneal injection of ketamine (50 mg/kg) plus xylazine (10 mg/kg), transcardially perfused with either the phosphate buffer saline (PBS) or artificial cerebrospinal fluid (ACSF) solution (see below), and sacrificed.

The induction and progression of the colon inflammation were evaluated by determining the disease activity index (DAI) and colon parameters, such as length, histological score, and mRNA relative expression of pro-inflammatory cytokines, as previously described [52]. For DAI evaluation, daily, throughout the DSS treatment, the animals were monitored for body weight, stool consistency, and presence of blood in feces. For histological scoring, paraffin-embedded sections of the distal colon were stained with hematoxylin/eosin and tissue damage was graded on destruction of epithelium, dilatation of crypts, loss of goblet cells, inflammatory cell infiltrate, edema, and crypt abscesses. Either for DAI or for histological assessment, each of the items were scored on a scale of 0 to 3 and added together to obtain a total severity score.

### 4.4. Biological Preparations 

For PCR and immunohistochemistry studies, control (untreated) and DSS-treated rats were anesthetized and transcardially perfused with PBS (in mM: 137 NaCl, 2.7 KCl, 10 Na_2_HPO_4_, and 1.8 KH_2_PO_4_; pH 7.4). The colons from the rats were removed, washed with ice-cold saline solution, and the weight and length were measured. Segments of distal colon were either fixed by overnight incubation with PBS containing 4% paraformaldehyde (PFA) for histological analysis or frozen in liquid nitrogen and stored at −80 °C until their use in RNA extraction to perform PCR measurements. 

For immunofluorescence assays, following the transcardial perfusion with PBS, the rats were perfused with 4% PFA in PBS. Thereafter, the brains were removed, post-fixed with PBS containing 4% PFA overnight, cryoprotected with sucrose solution in PBS, frozen in isopentane, and stored at −20 °C until use. Coronal sections of brain, including the primary motor cortex (20 μm-thick), were cut on a cryostat and applied to adhesive-coated glass slides.

For electrophysiology recordings, control and DSS-treated rats were transcardially perfused with artificial cerebrospinal fluid (ACSF), containing (in mM): 126 NaCl, 2 KCl, 1.25 NaH_2_PO_4_, 26 NaHCO_3_, 10 glucose, 2 MgCl_2_, and 2 CaCl_2_. The low-calcium ACSF solution contained 4 mM of MgCl_2_ and 0.1 mM of CaCl_2_. Both ACSF and low-calcium ACSF solutions were bubbled with 95% O_2_–5% CO_2_ (pH 7.4). Brains were removed and placed in low-calcium ACSF at 4 °C. Primary motor cortex slices (300 μm-thick) were obtained on a vibratome, kept at 33 °C for 30 min in an ACSF-filled chamber, and then stored at 21 °C in the same solution until use. For RNA extraction and PCR measurements, the primary motor cortex was dissected, frozen in liquid nitrogen, and stored at −80 °C until use. The primary motor cortex was identified as previously described [39].

### 4.5. Relative Quantification of Real-Time PCR

Real-time PCR was carried out as previously described [52]. Briefly, total RNA was extracted from both the distal colon and primary motor cortex of control and DSS-treated rats using the RNeasy^®^ kit (Qiagen, Madrid, Spain). cDNA was synthesized from 1 µg of total RNA using the QuantiTect^®^ reverse transcription kit (Qiagen), as described by the manufacturer. The primers used are presented in Table 3. Real-time PCR was performed with SsoFast™ EvaGreen^®^ Supermix (BioRad, Madrid, Spain). *β-Actin* served as a reference gene and was used for samples’ normalization. The comparative Ct method was applied to determine the mRNA relative expression and the value of each gene measured in control rats was set at 1.

### 4.6. Immunofluorescence Staining

Immunofluorescence assays were performed on coronal brain cryosections including the primary motor cortex of control and DSS-treated rats. Briefly, 20 μm-thick cryosections applied to adhesive-coated glass slides were washed with PBS, permeabilized with 1% Triton X-100 for 1 h, and blocked with 3% bovine serum albumin (BSA), 3% fetal calf serum (FCS), and 0.1% Triton X-100 in PBS for 1 h at room temperature. Sections were then incubated overnight at 4 °C with the primary antibodies anti-Iba1 and anti-iNOS or anti-GFAP diluted in blocking solution. Primary antibody binding was visualized with the appropriate secondary antibodies: Alexa Fluor-546 or -488. Nuclei were stained with Hoechst 33258 (Invitrogen, Thermo Fisher, Madrid, Spain). Negative controls for primary antibodies were performed in parallel without the correspondent primary antibody. The brain slides were mounted (Vectashield, Vector, Burlingame, CA, USA) and photographed with an Olympus BX61 microscope equipped with a DP73 camera. Images were analyzed using the ImageJ program. Immunofluorescence images were obtained at the “Centro de Investigación Tecnología e Innovación de la Universidad de Sevilla” (CITIUS).

### 4.7. Microglial and Astrocyte Activation

Markers for activated microglial cells (Iba1) and astrocytes (GFAP) were assessed by quantitative immunofluorescence. Increased expression of GFAP is used as a marker of astrocyte activation [53]. For activated microglia, we measured the Iba1 immunostaining area of both the microglia cell body and the whole cell to determine microglial morphological changes, following the method indicated by Hovens et al. [54], and the number of either Iba1- and/or iNOS-positive cells [55]. The analysis was performed by using the ImageJ program. The number of Iba1-positive cells (red color signal), iNOS-positive cells (green color signal), and Iba1-iNOS double staining positive cells (yellow-orange color signal) were counted choosing the same area in all sections (100,000 μm^2^). The results of morphological modifications are presented as microglial cell body size to total cell size ratio [54]. The number of immunofluorescent labelled cells referred to the analyzed area and was expressed as number of cells/mm^2^. For activated astrocytes, the GFAP immunostaining area was measured in a frame of known area (50,000 μm^2^) placed on the tissue section and expressed as the percentage of this area. We analyzed 2 immunolabelled sections/rat and in 5–8 fields/section covering a representative area of the primary motor cortex, and a mean of all measurements for each animal was calculated. Quantifications were performed by 3 independent observers blinded to the treatment. 

### 4.8. Whole-Cell Patch-Clamp Recordings and Analysis

Electrophysiological parameters were measured using the whole-cell patch-clamp technique and recordings in current- and voltage-clamp mode. The primary motor cortex slices were placed in the recording chamber, perfused with oxygenated ACSF with a pump (P-70, Harvard Apparatus, Cambridge, MA, USA) at 1 mL min^−1^, and warmed to 33 °C with a feedback-controlled heater (TC-324B, Warner Instruments Corporation, San Diego, CA, USA). Visual guidance to patch-clamp the neurons includes a Nikon Eclipse FN1 microscope with infrared-differential interference contrast (IR-DIC) optics, a 40× water-immersion objective, and a WAT-902H2 Ultimate Camera. Pyramidal neurons were identified based on their location within the motor cortex and their remarkable morphology [56]. Patch pipettes were pulled from a borosilicate glass capillary (inner diameter 0.6, outer diameter 1 mm; World Precision Instruments) using a glass microelectrode puller (PC10, Narishige, London, U.K). Patch electrodes had 3–5 MΩ resistance. The intracellular solution pipette was a K^+^-gluconate solution containing (in mM): 120 K^+^-gluconate, 10 KCl, 10 phosphocreatine disodium salt, 2 MgATP, 0.3 NaGTP, 0.1 EGTA, and 10 HEPES (pH 7.3). An osmolarity of 285 mosmol/kg was maintained via adjustment with sucrose [57].

Whole-cell patch-clamp recordings were achieved using a micromanipulator (MP-225, Sutter, Novato, CA, USA) and an amplifier (Multiclamp 700B, Axon Instruments, Molecular Devices, Sunnyvale, CA, USA). Recordings were low-pass Bessel-filtered at 2–10 kHz, and the data were digitized at 2–20 KHz with a Digidata 1550 analog–digital converter and developed on a computer screen using the pCLAMP 10 software (Molecular Devices, San Jose, CA, USA). Data were analyzed using the Clampfit 10.4 software (Molecular Devices, San Jose, CA, USA). In current-clamp mode, the bridge was periodically balanced using the auto-adjust feature. Throughout voltage-clamp recordings, the whole-cell capacitance was measured, and recordings were discontinued if the series resistance increased by >50% or exceeded 20 MΩ.

### 4.9. Current- and Voltage-Clamp Recordings

The following parameters were evaluated to assess the intrinsic electrical properties: resting membrane potential, input resistance, rheobase, voltage threshold, depolarization voltage, action potential amplitude and duration, frequency gain, maximum frequency of discharge, and cancellation current. The methods employed were as previously described [37]. The resting membrane potential was the difference between the intracellular and extracellular potentials after moving the electrode away. To calculate the input resistance, positive and negative current pulses were injected into the cell (500 ms, 1 Hz; 10 pA each step) and the slope of the current–voltage relationship affords the value of the resistance. Rheobase is the minimum current intensity (100 ms, 1 Hz; 5 pA each step) able to produce an action potential in 50% of cases. The voltage increment in membrane potential required to achieve the voltage threshold was the depolarization voltage. To determine the spike threshold, the action potential recording was differentiated, with the spike onset taken as the value of the membrane potential at which the first derivative exceeded 10 V s^−1^. Single action potential amplitude was the difference between the voltage at a resting level and that at the spike peak, and its duration was determined as the width of the spike at its half amplitude. Repetitive discharge was evoked by depolarizing current steps (1 s, 0.5 Hz; 10–50 pA increments). The firing frequency was taken as the number of spikes during the repetitive discharge in 1 s. The frequency gain was the slope of the firing frequency and current injected relationship. The maximum firing frequency was the highest frequency achieved by the neurons, regardless of the current intensity, and the cancellation current was the current at which the neuron firing ceases. 

### 4.10. Locomotor Activity and Motor Coordination Assessment

Motor behaviors were assessed in control and DSS-treated rats according to the established protocols [58,59].

The spontaneous locomotor activity was measured for 10 min in an active cage, consisting of a novel open-field arena (dimensions 45 × 45 cm) equipped with a photoelectric actimeter (LE8815-Panlab, Barcelona, Spain) with SEDACOM 2.0v software (Barcelona, Spain). The cage projects infrared light beams from one side of the cage to the other. Each time the rat moves around the cage, the beams are interrupted, and this is recorded as “beam break counts”. The arena and surrounding walls were cleaned with water and 70% of ethanol before each session. The different motor behavioral parameters were measured using ActiTrack v.2.7.13 software (PanLab, Barcelona, Spain) with the specific setup. The measurements included total and per minute horizontal motor activity (beam break counts), total distance traveled (in centimeters), locomotion in the center and in the periphery of the arena (beam break counts), percentage of time spent within the center and within the periphery of the arena, number of stereotypes (repetitive movements that is invariant and without a goal), number of grooming movements, total vertical motor activity (expressed in number of rearings), maximum speed (in centimeters/second), percentage of time spent resting, and slow locomotion and fast locomotion.

Motor coordination was evaluated using a Rotarod apparatus (LE8505-Panlab, Barcelona, Spain). We performed a two-trial test, with a maximum time of 5 min in the rotarod per trial, and the trials were separated by 20 min. Both trials were performed with a constant acceleration protocol from 4 to 40 rotations per minute (rpm). Rats were placed on the rotating rod, facing away from the direction of rotation so they must walk forward to stay upright, and the rotarod was set with a start speed of 4 rpm, and an acceleration rate of 20 rpm/min. Latency to fall: the time, in seconds, that an animal was able to hold itself on the rod, with a maximum time of 300 s, and maximum speed, in rpm, before the animal falls were recorded.

### 4.11. Statistical Analysis

Statistical analyses were conducted using GraphPad Prism v8.0 (San Diego, CA, USA) and IBM SPSS Statistics v26 programs (Armonk, NY, USA). Data are presented as mean ± standard error of the mean (SEM). To test normality, the Shapiro–Wilk test was applied. Comparisons between the two experimental groups (independent) were evaluated by the two-tailed Student’s *t*-test. When data were not normally distributed, the Mann–Whitney *U*-test was used. Comparisons between different measures in the same experimental group (dependent variables) were evaluated with a mixed ANOVA, considering the following assumptions: the dependent variable was measured on a continuous scale, it was normally distributed, and there was homogeneity of variances, and pairwise comparisons were corrected by the Bonferroni test. If the normality and/or the homogeneity of variances assumptions were violated, a Friedman test was used followed by a post hoc Wilcoxon signed-ranked test. Differences were set to be significant at *p* < 0.05, unless stated otherwise. 

## 5. Conclusions

This is the first report showing that DSS-induced acute colon inflammation produces microglial and astrocyte activation in the primary motor cortex. The microglial and astrocyte activation and the increased pro-inflammatory environment might trigger motor neuron hyperexcitability. This hyperexcitability is apparently mediated by a depolarized resting membrane potential, decreased rheobase, and increased action potentials’ frequency gain, that could make the pyramidal neurons more vulnerable to injury. Locomotion and motor coordination impairments were also detected in DSS-treated rats. As these alterations are present at the early stages of the ALS pathology, it could be inferred that colon inflammation contributes to ALS development.

## Figures and Tables

**Figure 1 ijms-23-05347-f001:**
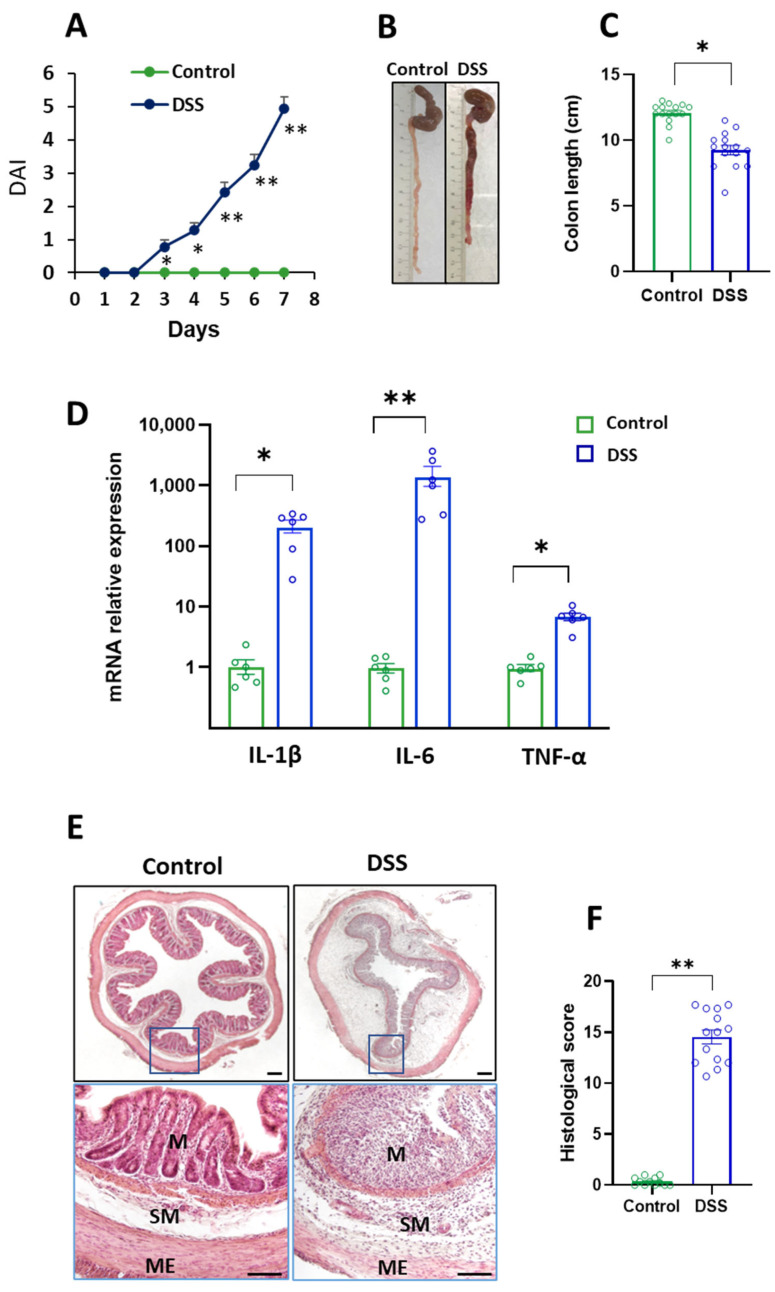
Evaluation of the DSS-induced acute colitis in rats. Control (untreated) and DSS-treated rats received either water or water containing 3% dextran sulphate sodium (DSS) for 7 days, respectively. (**A**) Disease activity index (DAI) assessed daily. (**B**) Representative macroscopic appearance of cecum and colon. (**C**) Colon length. (**D**) mRNA relative expression of pro-inflammatory cytokines in the distal colon. (**E**) Representative hematoxylin/eosin-stained sections of histological analysis. (**F**) Score of distal colon damage. M: mucosa, SM: submucosa, ME: muscularis externa. Scale bar represents 200 μm. Data are means ± SEM (*n* = 8–15 animals per group). Student’s *t*-test: * *p* < 0.05 and ** *p* < 0.01, DSS-treated rats vs. controls.

**Figure 2 ijms-23-05347-f002:**
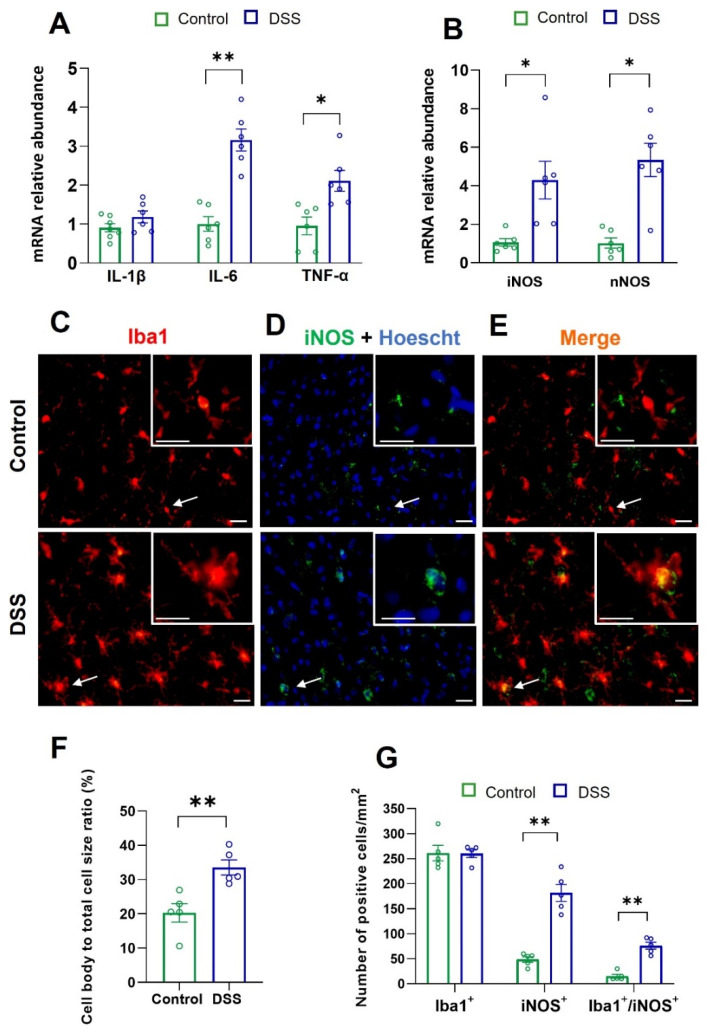
Neuroinflammation and microglial activation in the motor cortex of control and DSS-treated rats. mRNA relative expression of (**A**) pro-inflammatory cytokines and (**B**) iNOS and nNOS in the primary motor cortex (*n* = 6–8 animals per group). Immunolocalization of (**C**) Iba1 (red) and (**D**) iNOS (green) performed in brain sections containing the primary motor cortex. Nuclei were visualized with Hoechst (blue). (**E**) Colocalization of Iba1-iNOS (yellow-orange color). The cells pointed to by arrows are shown in higher magnification. Scale bar represents 20 µm. (**F**) Quantification of microglia cell body size (pixels) expressed as the cell body to cell size ratio (percentage) in the primary motor cortex. (**G**) Quantification of the number of either Iba1, iNOS, or Iba1-iNOS positive cells. For (**F**,**G**), the number of animals was 5 for each group and the number of cells analyzed in each animal was 200 (*n* = 5 animals per group). Data are means ± SEM. Student’s *t*-test: * *p* < 0.05 and ** *p* < 0.01, DSS-treated rats vs. controls.

**Figure 3 ijms-23-05347-f003:**
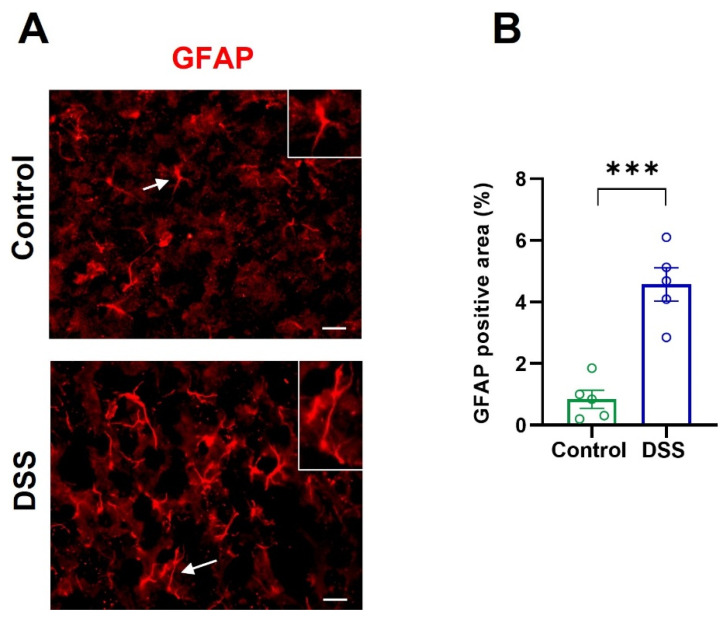
Astrocyte activation in the motor cortex of control and DSS-treated rats. (**A**) Immunofluorescence of GFAP was performed in brain sections containing primary motor cortex. The cells pointed to by arrows are shown in higher magnification. Scale bar represents 20 µm. (**B**) Quantification of GFAP immunostaining area expressed as percentage of the analyzed area in the primary motor cortex. The number of animals was 5 for each group and the number of measurements in each animal was 30–50 (*n* = 5 animals per group). Data are means ± SEM. Student’s *t*-test: *** *p* < 0.001, DSS-treated rats vs. controls.

**Figure 4 ijms-23-05347-f004:**
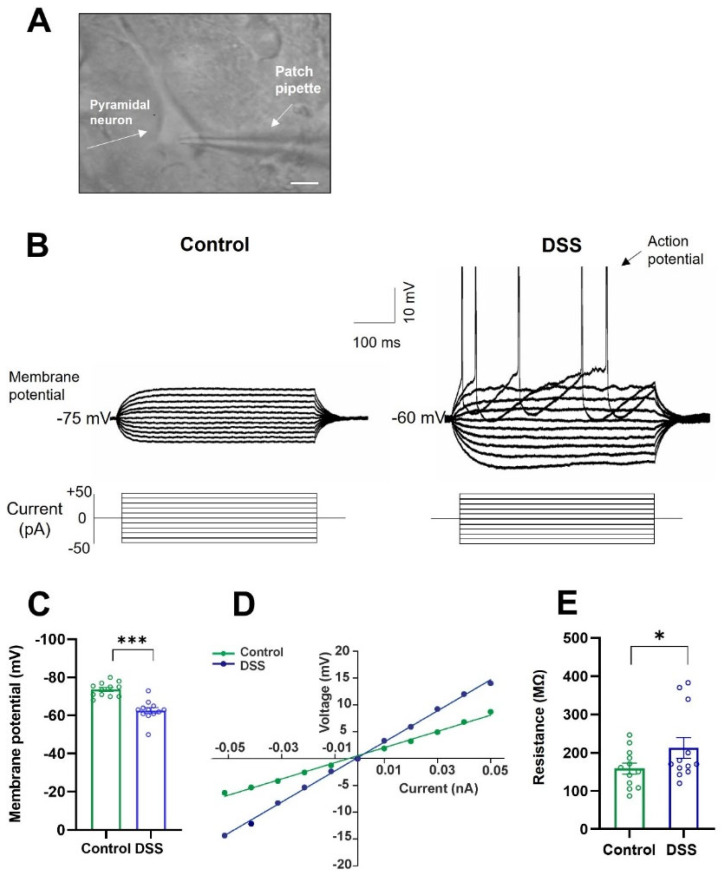
Membrane input resistance and resting membrane potential of pyramidal neurons from the motor cortex of control and DSS-treated rats. (**A**) Representative pyramidal neuron from the motor cortex, patch-clamped. Scale bar represents 20 µm. (**B**) Membrane voltage responses to depolarizing and hyperpolarizing current steps (500 ms, 10 pA) in a representative pyramidal neuron of control and DSS-treated rats. (**C**) Resting membrane potential (mV). (**D**) Relationship between values of current and voltage represented in (**B**). (**E**) Membrane input resistance (MΩ). Data are means ± SEM. The number of animals was 4 for each experimental condition and the number of neurons recorded from each animal was 2–4 (*n* = 12 cells recorded in total for each group). Student’s *t*-test: * *p* < 0.05 and *** *p* < 0.001, DSS-treated rats vs. controls.

**Figure 5 ijms-23-05347-f005:**
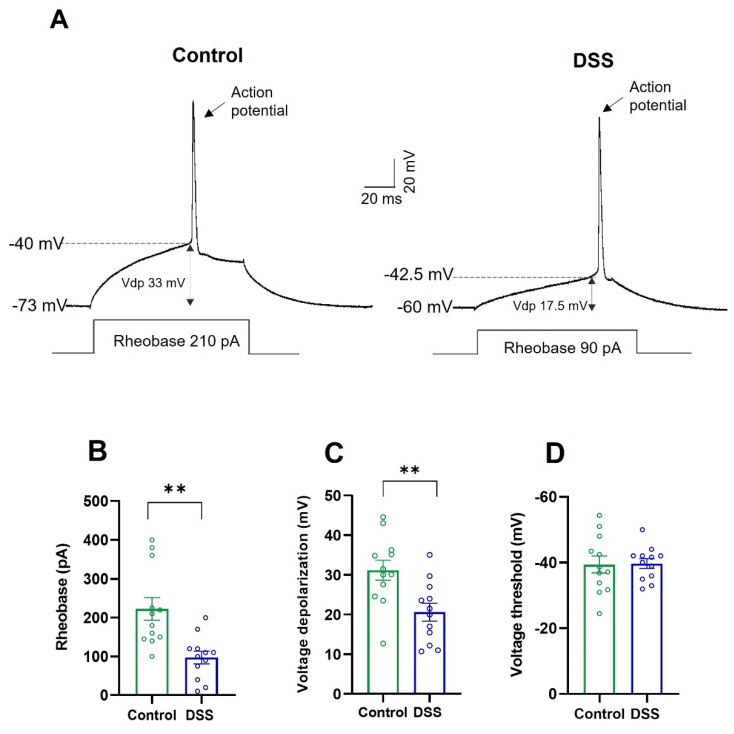
Rheobase and depolarization voltage of pyramidal neurons from the motor cortex of control and DSS-treated rats. (**A**) Membrane voltage responses to rheobase (minimum current required to evoke action potential) in a representative pyramidal neuron of control and DSS-treated rats. Vdp: Depolarization voltage. (**B**) Rheobase (pA). (**C**) Depolarization voltage (mV). (**D**) Voltage threshold (mV). Data are means ± SEM. The number of animals was 4 for each experimental condition and the number of neurons recorded from each animal was 2–4 (*n* = 12 cells recorded in total for each group). Student’s *t*-test: ** *p* < 0.01, DSS-treated rats vs. controls.

**Figure 6 ijms-23-05347-f006:**
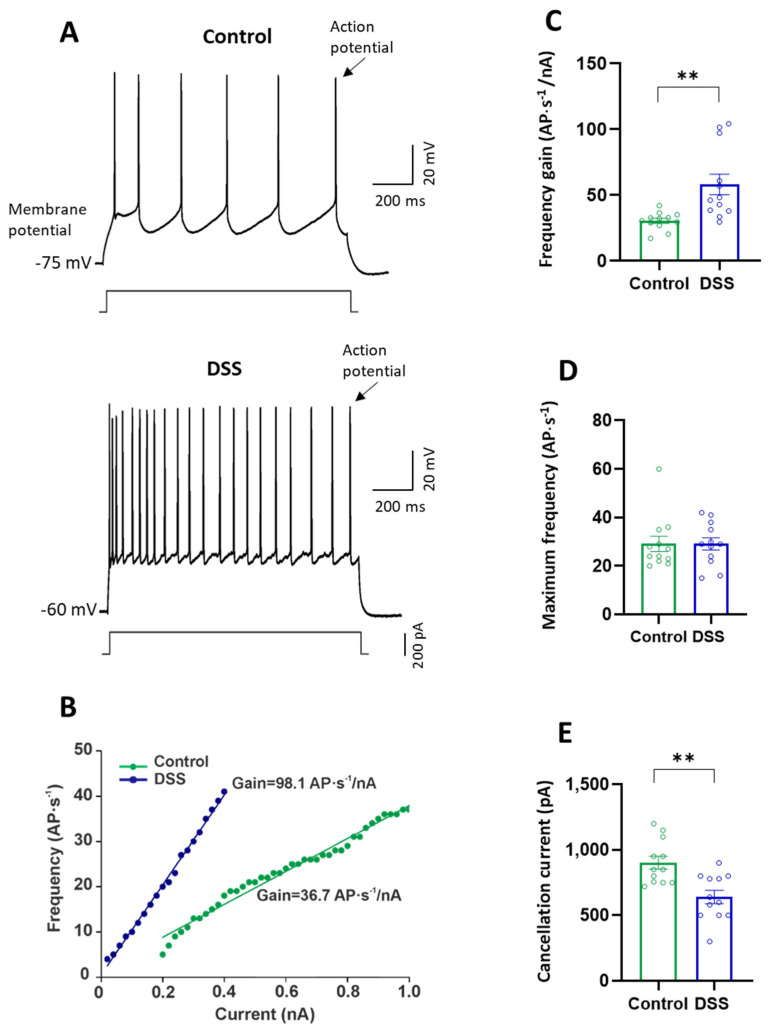
Firing properties of pyramidal neurons from the motor cortex of control and DSS-treated rats. (**A**) Membrane potential responses to long-lasting depolarizing current pulses (200 pA) in a representative neuron from control and DSS-treated rats. (**B**) Relationship between current intensity and frequency of action potentials (AP) represented in (A). (**C**) Frequency gain (AP·s^−1^/nA). (**D**) Maximum frequency (AP·s^−1^). (**E**) Cancellation current (pA). Data are means ± SEM. The number of animals was 4 for each experimental condition and the number of neurons recorded from each animal was 2–4 (*n* = 12 cells recorded in total for each group). Student’s *t*-test: ** *p* <0.01, DSS-treated rats vs. controls.

**Figure 7 ijms-23-05347-f007:**
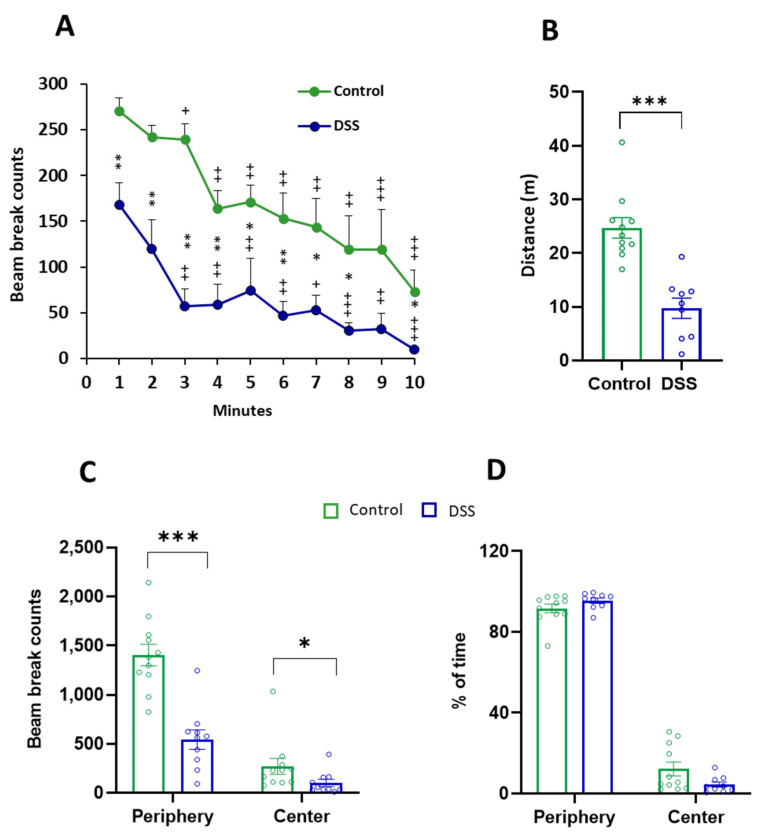
Horizontal motor activity in control and DSS-treated rats. Activity was measured in a 10 min period. (**A**) Horizontal activity per minute measured by the number of infrared beams broken by the rat (beam break counts). Friedman test showed that both groups were affected by the time factor (Control *p* < 0.0001; DSS-treated rats *p* < 0.0001). The Wilcoxon test was used for comparisons between different minutes in the same group: + *p* < 0.05, ++ *p* < 0.01, +++ *p* < 0.001, first minute vs. last minutes. (**B**) Total horizontal activity as distance travelled (m). (**C**) Locomotion in the center and in the periphery of the arena as beam break counts. (**D**) Percentage of time spent within the center and within the periphery of the arena. Data are means ± SEM (*n* = 10 animals per group). Student’s *t*-test or Mann–Whitney *U* test for comparisons between independent groups: * *p* < 0.05, ** *p* < 0.01, *** *p* < 0.001, DSS-treated rats vs. controls.

**Figure 8 ijms-23-05347-f008:**
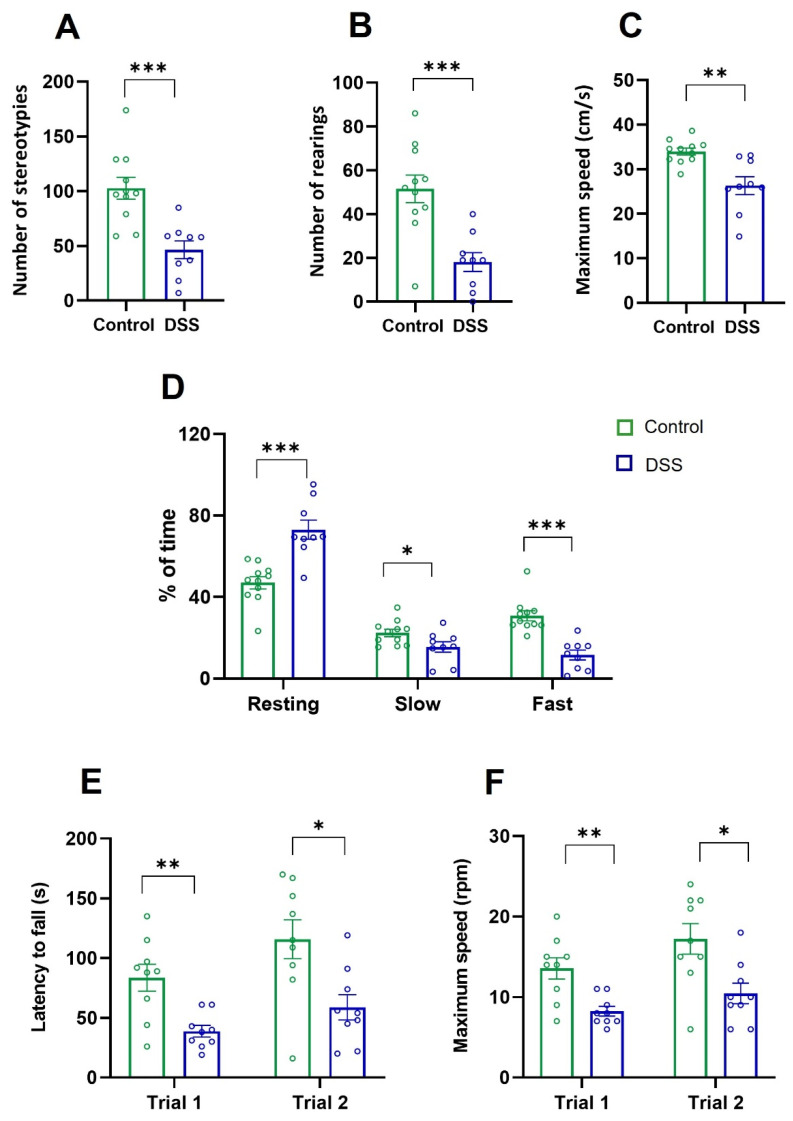
Vertical motor activity, locomotion speed, and rotarod test in control and DSS-treated rats. (**A**) Stereotyped movements. (**B**) Total vertical activity as number of rearings. (**C**) Maximum speed expressed as cm/s. (**D**) Percentage of time spent resting or performing either slow or fast movements. (**E**,**F**) Rotarod test: latency to fall (s) and maximum speed as rotations per minute (rpm). No significant intragroup differences were found between trials. Data are means ± SEM (*n* = 10 animals per group). Student’s *t*-test for comparisons between independent groups: * *p* < 0.05, ** *p* < 0.01, *** *p* < 0.001, DSS-treated rats vs. controls.

**Table 1 ijms-23-05347-t001:** Effects of DSS-induced acute colon inflammation on electrophysiological properties of pyramidal neurons from the primary motor cortex.

Membrane Properties	Control	DSS
Membrane resting potential (mV)	−73.66 ± 1.06	−62.65 ± 1.52 ***
Input resistance (MΩ)	158.49 ± 14.35	212.6 ± 27.14 *
Rheobase (pA)	222.5 ± 4.94	97.08 ± 16.19 **
Voltage depolarization (mV)	31.16 ± 2.50	20.60 ± 2.22 **
Voltage threshold (mV)	−39.42 ± 2.54	−39.71 ± 1.48
Action potential amplitude (mV)	120.38 ± 4.34	112.95 ± 3.21
Action potential duration (ms)	1.69 ± 0.07	1.57 ± 0.08
Frequency gain (AP·s^−1^/nA)	30.21 ± 7.89	57.93 ± 7.89 **
Maximum frequency (AP·s^−1^)	29.08 ± 3.17	29.10 ± 2.56
Cancellation current (pA)	902.16 ± 48.5	640. 00 ± 50.57 **

Data are presented as mean ± standard error of the mean. The number of animals was 4 for each experimental condition and the number of neurons recorded from each animal was 2–4 (*n* = 12 cells recorded in total for each group). Student’s *t*-test: *** *p* < 0.001, ** *p* < 0.01, * *p* < 0.05, DSS-treated vs. control rats.

**Table 2 ijms-23-05347-t002:** Effects of DSS-induced acute colon inflammation on locomotion and motor coordination.

Motor Behaviors	Control	DSS
Total horizontal activity (beam break counts)	1672.2 ± 157.0	664.5 ± 134.4 ***
Total distance travelled (m)	24.73 ± 1.9	9.76 ± 1.8 ***
Locomotion (beam break counts)		
Periphery	1309.9 ± 106.5	524.3 ± 102.9 ***
Center	259.5 ± 80.5	105.7 ± 39.7 *
Percentage of time spent (%)		
Periphery	91.6 ± 2.1	95.4 ± 1.3
Center	12.2 ± 3.4	4.5 ± 1.3
Stereotyped movements (number)	102.7 ± 9.9	46.4 ± 8.1 ***
Total vertical activity (number of rearings)	56.0 ± 4.9	18.1 ± 4.2 ***
Maximum speed (cm/s)	33.9 ± 0.8	1.6 ± 0.3 **
Percentage of time (%)		
Resting	46.9 ± 2.9	73.1 ± 4.6 ***
Slow locomotion	22.3 ± 1.8	15.4 ± 2.5 *
Fast locomotion	30.7 ± 2.5	11.4 ± 2.4 ***
Latency to fall (s)		
Trial 1	79.6 ± 11.9	38.7 ± 4.8 **
Trial 2	113.1 ± 18.14	58.8 ± 10.6 *
Maximum speed in rotarod test (rpm)		
Trial 1	13.1 ± 1.4	8.2 ± 0.6 **
Trial 2	16.7 ± 2.0	10.4 ± 1.2 *

Data are presented as mean ± standard error of the mean. The number of animals was 10 for each experimental condition (*n* = 10 for each group). Student’s *t*-test or Mann–Whitney *U* test: * *p* < 0.05, ** *p* < 0.01, *** *p* < 0.001, DSS-treated vs. control rats.

**Table 3 ijms-23-05347-t003:** Oligonucleotides sequences used for reverse transcription-polymerase chain reaction assays.

Gene Symbol	Accession No.	Sense (5′-3′)	Antisense (5′-3′)
** *IL-1β* **	NM_031512.2	CTTTCGACAGTGAGGAGAATGAC	CCACAGCCACAATGAGTGAC
** *IL-6* **	NM_012589.2	ACAAGTCGGAGGCTTAATTACA	GAAAAGAGTTGTGCAATGGCAA
** *iNOS* **	NM_012611.3	GATGTTGAACTACGTCCTATCTCC	GTCTTG GTG AAAGCGGTGTTC
** *nNOS* **	NM_052799.1	CCTTTGAATACCAGCCTGATCC	TTGTGATTTGCCTGTCTCTGTG
** *TNF-α* **	NM_012675	CTCACACTCAGATCATCTTCTC	TGGTATGAAATGGCAAATCGG
** *β-actin* **	NM_007393	ACCCACACTGTGCCCATCTA	CGGAACCGCTCATTGCC

Primers were chosen according to the cDNA sequences entered in GenBank and designed using PerlPrimer program v1.1.14 (Parkville, Victoria, Australia). *IL-1β*, interleukin 1β; *IL-6*, interleukin 6; *iNOS*, inducible nitric oxide synthase; *nNOS*, neuronal nitric oxide synthase; *TNF-α*, tumor necrosis factor α.

## Data Availability

The datasets used and/or analyzed during the current study are available from the corresponding author upon reasonable request.

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
