# Peer review of "Acute Colon Inflammation Triggers Primary Motor Cortex Glial Activation, Neuroinflammation, Neuronal Hyperexcitability, and Motor Coordination Deficits"

_ijms, 2022, doi:10.3390/ijms23105347_

Round 1

Reviewer 1 Report

Carrascal L et al investigated the impact of acute colitis on brain function, and in particular on the primary motor cortex, involved in motion execution, in rats. Acute colonic inflammation induced by short term DSS treatment resulted in neuroinflammation and neuronal hyperexcitability, which was associated with impaired motor activity. This work is in line with the attractive hypothesis of a gut contribution to the development of brain disorder, including neurodegenerative diseases, notably through a deletorious gut/brain inflammatory axis. This work is original as it is focused on a brain region that has not been extensively studied in this context. It may thus have an important impact for neurodegenerative diseases affecting this brain area such as amyotrophic lateral sclerosis, with still poorly characterized aetiology. The paper is well writen. Research methodology is logical and globally appropriate. Here are my specific comments :

1) The authors show that DSS induces microglia activation, by studying morphological changes and iNOS expression of Iba1+ cells. Although Iba1 is a widely used marker to stain microglia, it is not specific to this cell type and is shared with non brain-resident macrophages, that may infiltrate the brain under inflammatory conditions. The authors should use microglia-specific markers or at least add a comment on this point.

2) Microglia activation was not linked to any change in microglia absolute number. However the methodology to monitor microglia absolute number was not clear. Indeed, the authors present a % among total cells but one can not exclude that DSS may induce a decrease in other cells (neurons for example) which may compensate for a possible increase in microglia number. Do the authors have more accurate data on microglia absolute number in this context ? More generally, the methodology of cells quantification should be described more accurately. Hence the number of cells analysed for each animal was « 10 ». Is this enough to calculate a % (Fig 2G) ?

3) Microglia can play both a cytotoxic and protective role in the brain, depending on the context and on their activation phenotype, oversimplified into M1 and M2 states. Alhtough M1-activated iNOS+ microglia can be cytotoxic, M2 microglia play a role in tissue protection following acute injury. Have the authors study some M2 markers, such as CD206 or Arg1 in the context of acute colitis ?

4) Microglia are main players in central nervous system inflammation. However, infiltrating immune cells, such as neutrophils, monocytes or T cells are also important, especially in the context of blood-brain barrier disruption such as in colitis. Do the authors have data on immune cells infiltrating the primary motor cortex following colitis ?

5) Do the authors have data on potential neuronal cytotoxicity ?

6) Colitis induces anxiety, visceral and systemic pain, weight loss, potential nutritional stress, which can all be confounding factors in the behavioral tests decribed in the study. How do the authors interpret their data in light of these colitis-associated confounding factors ? Did the authors use other tests of motor functions than the rotarod to confirm their findings ? This should be discussed and data interpreted with caution.
Similarly how do the authors reconcile their finding on the lack of anxiety induced by DSS with the literature showing that colitis results in anxiety-like behavior ?

7) What was the rationale for choosing male rats ?

Minor comments :
-Fig 2G is not clear. The colors used and the legend may be revised.
-In the Materials and Methods, some references are missing (antibodies for immunofluorescence staining)

Author Response

Reviewer #1:

Carrascal L et al investigated the impact of acute colitis on brain function, and in particular on the primary motor cortex, involved in motion execution, in rats. Acute colonic inflammation induced by short term DSS treatment resulted in neuroinflammation and neuronal hyperexcitability, which was associated with impaired motor activity. This work is in line with the attractive hypothesis of a gut contribution to the development of brain disorder, including neurodegenerative diseases, notably through a deletorious gut/brain inflammatory axis. This work is original as it is focused on a brain region that has not been extensively studied in this context. It may thus have an important impact for neurodegenerative diseases affecting this brain area such as amyotrophic lateral sclerosis, with still poorly characterized a etiology. The paper is well written. Research methodology is logical and globally appropriate. Here are my specific comments:

1) The authors show that DSS induces microglia activation, by studying morphological changes and iNOS expression of Iba1+ cells. Although Iba1 is a widely used marker to stain microglia, it is not specific to this cell type and is shared with non brain-resident macrophages, that may infiltrate the brain under inflammatory conditions. The authors should use microglia-specific markers or at least add a comment on this point.

Answer: We agree that Iba1 staining cannot distinguish resident from peripheral microglia, this distinction is difficult because they share a majority of markers as discussed in this recent review Jurga et al. 2020, doi: 10.3389/fncel.2020.00198. In addition, some of the markers that are specific for resident microglia however detect microglia in resting state. In our study, part of the detected microglia may be peripheral cells. Nonetheless, although we agree that it would be a very interesting idea to be developed in future studies, it was not the main aim of the work to deeply study the phenotype of these cells. In light of this comment, we have added a sentence modifying the interpretation of our results in the revised manuscript (see lines 354-355).

2) Microglia activation was not linked to any change in microglia absolute number. However the methodology to monitor microglia absolute number was not clear. Indeed, the authors present a % among total cells but one can not exclude that DSS may induce a decrease in other cells (neurons for example) which may compensate for a possible increase in microglia number. Do the authors have more accurate data on microglia absolute number in this context ?

Answer: We agree that the number of cells can change although we did not find a significant difference in the % of total cells between control and DSS-treated rats. We have more accurate data and we show them in the revised manuscript expressed as number of positive cells per area (see lines 146-150 and 550-552).

More generally, the methodology of cells quantification should be described more accurately.

Answer: We have described more accurately this methodology in the revised manuscript (see lines 540-563).

Hence the number of cells analysed for each animal was « 10 ». Is this enough to calculate a % (Fig 2G) ?

Answer: We have made an honest mistake indicating the number of cells analyzed in figure 2G, the number of analyzed cells per animal was 200. We have corrected this in the revised manuscript (see Figure 2 legend).

3) Microglia can play both a cytotoxic and protective role in the brain, depending on the context and on their activation phenotype, oversimplified into M1 and M2 states. Alhtough M1-activated iNOS+ microglia can be cytotoxic, M2 microglia play a role in tissue protection following acute injury. Have the authors study some M2 markers, such as CD206 or Arg1 in the context of acute colitis ?

Answer: Although traditionally, macrophage and microglial activation has been classified in M1 and M2, this differentiation is controversial and nowadays does not respond to the variety of microglial phenotype found in the brain and instead a broad array of activation phenotypes has been recently described. In addition, microglia activation profile seems to be region and context dependent in brain disorders (see review Bachiller et al. 2018; DOI: 10.3389/fncel.2018.00488) and it would be very attractive to focus on these phenotypes in further studies.

4) Microglia are main players in central nervous system inflammation. However, infiltrating immune cells, such as neutrophils, monocytes or T cells are also important, especially in the context of blood-brain barrier disruption such as in colitis. Do the authors have data on immune cells infiltrating the primary motor cortex following colitis ?

Answer: Based in the reviewer´s question, we have performed immunofluorescence assays to detect some markers of immune cells with the antibodies such as anti-Cd4 and anti-Cd8 and we did not find positive immune staining. We cannot rule out whether other types of infiltrating immune cells could be found, but it will be an interesting goal to pursue in further studies.

5) Do the authors have data on potential neuronal cytotoxicity ?

Answer: We agree with the reviewer on the possibility of neuronal cytotoxicity since the observed increase in nNOS mRNA could indicate that the produced NO combined with superoxide forms the free radical peroxynitrite thus inducing cell death. However, we have analyzed caspase 3/7 activity by using the Caspase-Glo 3/7 Assay (Promega) and we did not find significant differences between both experimental group (100 ± 11.4% in controls vs. 104.9 ± 15.6% in DSS treated rats, n= 6 animals per group).

6) Colitis induces anxiety, visceral and systemic pain, weight loss, potential nutritional stress, which can all be confounding factors in the behavioral tests described in the study. How do the authors interpret their data in light of these colitis-associated confounding factors?

Answer: In our study the DSS treatment did not induce anxiety-like behavior, since we did not find significant differences between control and DSS-treated rats in the time spent in the center of the cage. In addition, the day of the behavior test, DSS-treated rats did not display signs of pain such as abnormal gait or posture, decreased grooming and muscular strength, and excessive diminution on body weight gain. This indicates that it is unlikely that the observed motor impairments in DSS-treated rats were due to sickness behaviors. We have included these data in a supplementary table in the revised manuscript and have made some comments about this in the discussion (see lines 301-306 and and 427-429).

Did the authors use other tests of motor functions than the rotarod to confirm their findings? This should be discussed and data interpreted with caution.

Answer: We have not used other specific tests than rotarod to show motor coordination. However, our results obtained from open field test by using the active cage revealed changes in motor behaviors that require coordination as vertical activity. We have discussed these data more cautiously in the revised manuscript (see lines 69-70 and 436).

Similarly how do the authors reconcile their finding on the lack of anxiety induced by DSS with the literature showing that colitis results in anxiety-like behavior ?

Answer: In other studies, treatments and animal models were different, e.g. Chen et al. 2015; 10.1152/ajpregu.00298.2014 found anxiety-like behavior but the % of DSS was higher thus inducing a more severe colitis. We have included comments related to this in the revised manuscript (see lines 430-432).

7) What was the rationale for choosing male rats ?

Answer: Estrogens have anti-inflammatory activity in brain. Studies have shown an inhibitory activity of estrogens on neuroinflammation and specifically on microglia (Vegeto et al., 2008; doi: 10.1016/j.yfrne.2008.04.00; Villa et al., 2018; doi: 10.1016/j.celrep.2018.05.048). It will be interesting to compare males with females, and it will be a further aim of the study in our laboratory.

 Minor comments:

 -Fig 2G is not clear. The colors used and the legend may be revised.

Answer: We have changed colors and legends in the figure 2G to make it clearer.

-In the Materials and Methods, some references are missing (antibodies for immunofluorescence staining)

Answer: We have completed the references of antibodies used in the immunofluorescence assays (see line 453).

Reviewer 2 Report

The paper entiteled „Acute colon inflammation triggers primary motorcortex microglial activation, neuroinflammation, neuronal hyperexcitability and motor coordination” by Livia Carrascal et al., investigates in a rat model of colitis ulcera how colonic inflammation affects the primary motor cortex. Thus, first the authors assessed microglial activation and the inflammatory response in the cortex of the lesioned rats. Neuronal function was monitored by electrophysiology, and behavioral tests. The detected hyperexcitablity of cortical neurons was assumed to be based on the colonic inflammation and to lead to motor deficits what might shed insights into human pathologies.

This is an interesting study that deserves attention, however, several concerns arose.

  1. The study lacks mechanistic insights. The finding that pro-inflammatory cytokines are induced in the brain of rodents with intestinal/colonic inflammation is not new. Thus the study would be improved massively if a mechanism of inflammatory action would be included, e.g. the induction of the immediate early gene c-fos could be assessed and associated with the cortical regions responsible for hyperexcitation and motor deficits.

  1. The statistics in Fig. 2 (and other figures) are not exactly clear. If 10 measurements per animal were performed, a mean should be calculated for each animal and the resulting n of 5 should be subjected to statistical analysis. This must be clarified.

  1. Activation of glial cells should be assessed

  1. The presence and effect of immune cells should be assessed.

  1. The study lacks prove in humans.

Author Response

Reviewer #2:

The paper entiteled „Acute colon inflammation triggers primary motorcortex microglial activation, neuroinflammation, neuronal hyperexcitability and motor coordination” by Livia Carrascal et al., investigates in a rat model of colitis ulcera how colonic inflammation affects the primary motor cortex. Thus, first the authors assessed microglial activation and the inflammatory response in the cortex of the lesioned rats. Neuronal function was monitored by electrophysiology, and behavioral tests. The detected hyperexcitablity of cortical neurons was assumed to be based on the colonic inflammation and to lead to motor deficits what might shed insights into human pathologies.

This is an interesting study that deserves attention, however, several concerns arose.

  1. The study lacks mechanistic insights. The finding that pro-inflammatory cytokines are induced in the brain of rodents with intestinal/colonic inflammation is not new. Thus, the study would be improved massively if a mechanism of inflammatory action would be included, e.g., the induction of the immediate early gene c-fos could be assessed and associated with the cortical regions responsible for hyperexcitation and motor deficits.

Answer: We have followed this recommendation and we have performed PCR assays to determine mRNA relative abundance of c-Fos but we have not observed differences between control and DSS groups (100 ± 13.5% in controls vs. 84 ± 27.1% in DSS-treated rats, n=6 animals per group). According with this we have not found either c-Fos positive staining in the immunofluorescence assays.

  1. The statistics in Fig. 2 (and other figures) are not exactly clear. If 10 measurements per animal were performed, a mean should be calculated for each animal and the resulting n of 5 should be subjected to statistical analysis. This must be clarified.

Answer: We have changed the figures 2F and G to make it clearer. We have made a mistake indicating the number of measurements, then the number of analyzed cells per animal was 200. We have reviewed the statistics calculating the means for each animal and then in this case the total number of animals used per experimental group was 5 (n=5). In the figures 1, 7 and 8 each measurement corresponds to an animal and the figures 4, 5 and 6 that correspond to electrophysiology experiments, results obtained from each neuron are represented as independent data.

  1. Activation of glial cells should be assessed

Answer: We have assessed activation of astrocytes by immunofluorescence assays to detect of glial fibrillary acidic protein (GFAP) used as marker. We have evaluated these glial cells because they are also involved in neuroinflammation. We have found a significant increase in the GFAP immunostaining in DSS-treated rats as compared with controls indicating activation. We have included these results (figure 3) in the revised manuscript (see lines 142-145 and 315-317).

  1. The presence and effect of immune cells should be assessed.

Answer: Based in the reviewer´s suggestion, we have performed immunofluorescence assays for some markers of immune cells with the antibodies such as anti-Cd4 and anti-Cd8 and we did not find positive immune staining. We cannot rule out whether other types of infiltrating immune cells could be found, and it will be interesting for further studies.

  1. The study lacks prove in humans.

Answer: It will be very interesting that in a near future this study will be perform using human samples, however it was our first goal to focus on a murine model.

Round 2

Reviewer 2 Report

No further comments.